# Nanopore sequencing reveals hidden landscape of short L1 transductions in colorectal cancer
Päivi Nummi [1,2], Aurora Taira [1,2], Janne Ravantti [1,2,3], Tuukka Norri [1,2,4], Niko Välimäki [1,2,5], Anna Lepistö[6], Laura Renkonen-Sinisalo[6], Selja Koskensalo [6], Toni T. Seppälä [1,5,7,8,9], Ari Ristimäki [1,10], Kyösti Tahkola [11], Anne Mattila[11], Jan Böhm[11], Jukka-Pekka Mecklin[12,13], Emma Siili[10], Annukka Pasanen[10], Oskari Heikinheimo [14], Ralf Bützow [10,14], Auli Karhu [1,2], Lauri A. Aaltonen [1,2,5], Kimmo Palin [1,2,5] ✉ & Tatiana Cajuso [15,16] ✉

L1s are repetitive sequences capable of copying themselves into new genomic loci. While L1s are typically repressed by DNA methylation in somatic tissues, they can become reactivated in cancer. Although L1 sequences are highly repetitive, ~25% of insertions carry a unique downstream sequence, transduction, that can be used to trace the source L1. Here, we apply nanopore long-read sequencing to 56 colorectal cancer samples to comprehensively detect somatic transductions and to characterize the source L1 activity. We demonstrate that earlier methods systematically miss a large proportion of mostly shorter transductions, leading to an incomplete and biased view of source L1 activity. Our analysis reveals a strong positive correlation between the number of transductions and other L1 insertions within samples and that distinct source L1s exhibit varying transduction lengths and 5' inversion frequency. Finally, we integrate DNA methylation provided by nanopore reads and show that active elements in cancer samples have lower methylation levels in contrast to inactive L1s. Together, our results provide a more complete characterization of somatically active L1 elements in colorectal cancer and highlight the utility of long-read sequencing in retrotransposon research.

Long interspersed nuclear element-1s (LINE1 or L1s) are retrotransposons, repetitive genetic elements that have the ability to create copies of themselves and insert into new loci in the genome. L1s remain the only active autonomous retrotransposons in the human genome, meaning they are not dependent on proteins from other retrotransposons. Retrotransposons, including L1s, mobilize via messenger-RNA (mRNA) intermediate in a mechanism known as copy-and-paste[1].

Approximately 17% of the human genome is composed of L1 elements, however not all of them remain retrotransposition competent[2]. Of the estimated 500,000 L1 elements in the human genome, only ~100 human specific L1s (L1HS) have preserved their capability to retrotranspose[3–5]. The majority of the L1 elements are truncated leaving them incompetent for transposition, as it is only possible for full-length elements (6 kbp) with 2 intact open reading frames[6].

New copies of L1s emerge from target primed reverse transcription (TPRT)[7], where the mRNA tail hybridizes with a target site polyT tract, which functions as a primer for reverse transcription (RT). As a result of TPRT, the insertion contains the hallmarks of retrotransposition: target site

duplication, endonuclease cut site, and a polyA-tail. In TPRT, RT initiates from the 3' end of the L1 sequence often leading to a truncated product with only the 3' end of the element and inability to mobilize further. Furthermore, the insertion can be truncated by a 5' inversion, which arises from a poorly understood mechanism known as twin priming[8].

Approximately 25% of L1 insertions contain sequence beyond the 3' end of the source L1[9]. These insertions, 3' transductions occur when downstream genomic sequence gets transcribed as L1 transcription machinery bypasses the polyadenylation signal at the 3' end of L1 elements[10]. In most cases, the canonical polyadenylation signal (AATAAA), or a variation of it, stops the transcription 10–30 bp downstream[11]. As RT starts from the 3' of the L1 DNA, the transduced sequence is the first to be reverse transcribed after the polyA-tail. Based on the inserted sequence content, transductions can be divided into two categories: partnered transductions which contain both L1 sequence and 3' downstream genomic sequence, and orphan transductions, which include only the 3' downstream genomic sequence. In both cases, the unique sequence outside the element can be used as a fingerprint to identify the source L1 producing the novel

---

**Fig. 1 | Landscape of L1 retrotransposition and transduction activity in colorectal cancer. a** A scatterplot showcasing the correlation between the number of somatic solo-L1 insertions and the number of transductions in 56 CRC samples. **b** Distribution of somatic transductions by source L1 across CRC samples, where individual bars correspond to individual samples. Source L1s are ordered by frequency in the legend. L1s producing less than 10 insertions are part of the group "Other". **c** Source L1s activating in both germline and in soma. Bar length indicates the total number of transductions per element separated by germline and somatic status.

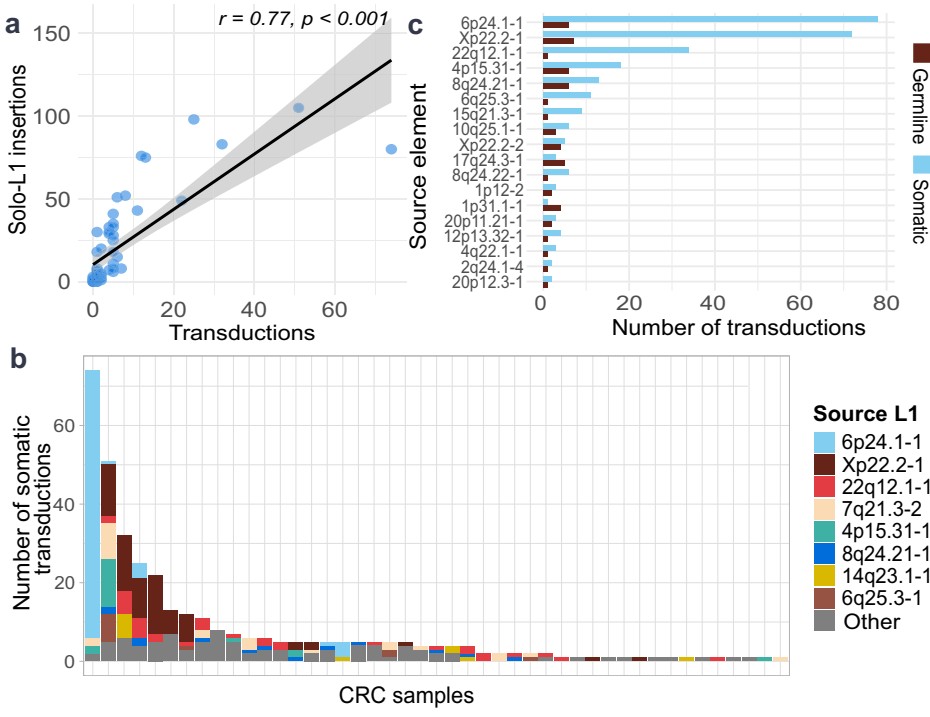

insertions[12]. This is not possible with solo-L1 insertions containing only L1 repetitive sequence.

L1 elements are silenced in most adult somatic tissues by a key epigenetic mechanism, promoter methylation, contributing to the suppression of their transcriptional activity[13]. However, low levels of L1 activity have been detected in germ cells and early embryonic development[14,15], healthy colon[16], and brain[17]. In addition to healthy somatic tissues, novel L1 insertions occur in many cancers where genome-wide hypomethylation, a hallmark of cancer, can lead to L1 promoter deregulation[18,19]. While promoter demethylation is insufficient to drive L1 expression[20], it is thought to be the main switch for L1 transcription[16].

Colorectal cancer (CRC) is globally the third most common cancer leading to almost 1 million cancer deaths annually[21]. Colorectal tumors exhibit some of the highest levels of somatic L1 retrotransposition among human cancers[22]. The role of L1 activity in cancer remains incompletely understood as many insertions occur within introns of genes not implicated in tumorigenesis[23]. However, somatic L1 insertions have, in rare cases, been shown to truncate the tumor suppressor gene *APC*, contributing to the initiation of colorectal cancer[24–26]. Moreover, high L1 activity has been associated with poor survival in CRC[25].

Previous studies of somatic L1 insertions in cancer have leveraged 3′ transductions to identify active source L1s[9,16,22,27]. These studies have consistently shown that a small number of highly active "hot" L1s account for the majority of observed transductions, with an L1 in chromosome 22q12.1 recognized in all of them. It is an example of a *Strombolian* element, L1 that is frequently active but producing few insertions per tumor, contrasted by *Plinian* elements that are rarely active but generating numerous insertions when activated[22]. Efforts to identify and characterize these active L1s have traditionally relied on retrotransposition assays in cultured cells[4,5] and short-read sequencing approaches[9,16,22,25]. However, these methods are limited in their ability to resolve insertion sequences.

In this study, we apply nanopore long-read sequencing to investigate L1 retrotransposition in colorectal cancer. Long sequencing reads allow us to identify active source elements and perform a detailed analysis on the transduction sequences and explore characteristics of different source L1 elements in CRC. Furthermore, we integrate information on DNA methylation levels of the source L1 elements and study the link between

methylation and transduction activity in tissues with and without transduction activity. In total, our results provide a detailed catalog of L1 activity in colorectal samples accompanied by the source element methylation.

## Results
### High-resolution mapping of somatic L1 transductions in colorectal cancer

To characterize the landscape of active L1 elements in colorectal cancer (CRC), we performed long-read Oxford Nanopore sequencing and applied TraDetIONS[28] on 56 primary colorectal tumors and 12 matched normal colon tissues. This approach enabled an estimated precision of 93–96% detection of somatic L1 insertions[28]. To detect the active source L1s from the resulting somatic insertion set, we focused on 3′ transductions. The transductions were identified by utilizing reconstructed insertion sequences mapping within 3 kb of candidate source L1s (Methods: TE and transduction detection). Out of all the detected somatic L1 insertions, 349/1531 (23%) carried traceable 3′ transductions, allowing confident assignment to their source loci. All somatic transductions detected originated from a total of 43 distinct full-length L1HS elements (Supplementary Data S1 and S2). The number of transductions per tumor sample strongly correlated with the total number of somatic L1 insertions per sample (Pearsons, $r = 0.77$, $p = 2.7 \times 10^{-12}$), indicating that transduction burden can serve as a proxy for overall retrotransposition activity (Fig. 1a and Supplementary Data S3).

In CRC samples with more than two detected transductions, multiple source L1s contributed to the overall insertion burden with some tumors exhibiting transduction activity up to 12 distinct sources (Supplementary Data S2 and S3 and Fig. 1b). The activity of the source L1s varied in tumors, for example, the fixed L1 22q12.1-1 was active in 17 tumors but produced only a small number of insertions per tumor, while the polymorphic L1 in 6p24.1-1 was active in just four tumors yet generated 68 insertions in a single sample—the highest number of insertions in our cohort. The polymorphic L1 in Xp22.2-1 showed an intermediate behavior, contributing ≥10 insertions in several tumors. This variability in both number of transductions per sample and number of samples where active resembles the *Strombolian* and *Plinian* patterns of L1 behavior described previously[22], where some loci are frequently but modestly active, and others are rarely active but capable of producing large insertion bursts when active. However, most of the sources

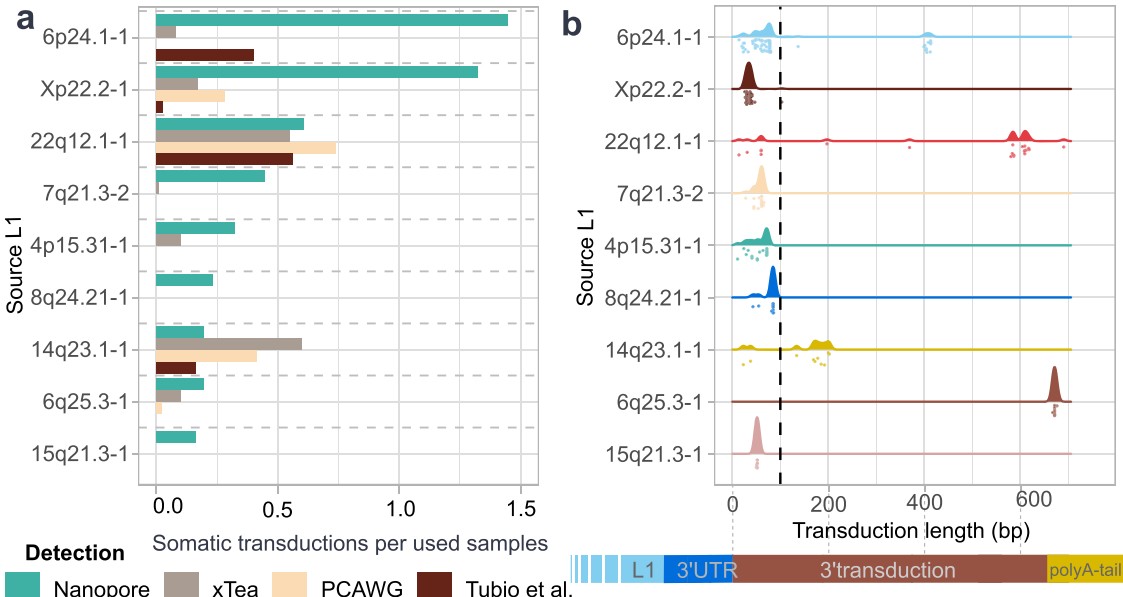

**Fig. 2 | Comparison to prior detection with transduction length. a** Relative activity of the most somatically active source L1s in nanopore sequenced CRC tumors, compared to datasets sequenced using Illumina technology and analyzed with xTea[29] or TraFiC-mem[9,22]. Relative activity is expressed as the number of somatic transductions divided by number of samples used in the detection. **b** Length distribution of 3′ transductions for the source L1s shown in (**a**). Nanopore sequencing enables detection of short transductions that are frequently missed by short-read platforms.

in our data do not follow this bimodal model, but rather fall in between the two extremes, suggesting a spectrum rather than binary classification. Our findings reinforce the idea that somatic retrotransposition is typically driven by a small, variably active subset of L1HS elements.

As expected, the detected somatic transductions in this study exhibited hallmarks of retrotransposition, over 99% of transductions displayed at least one hallmark (348/349) (Supplementary Data S4). Furthermore, partnered transductions with both L1 and transduction sequence are often characterized by two polyA tails: one from the original source L1 and the other from the novel transduction (Supplementary Fig. S1). Of the 217 partnered transductions, 72% (215/297) presented two polyA-tails within the inserted sequence (Supplementary Data S4).

In addition to somatic transductions, we identified 153 germline transductions arising from 70 source L1s. Eighteen of the somatically active source L1s were also active in the germline (Fig. 1c). These shared elements accounted for 80% of all somatic insertions but only 30% of germline insertions. This suggests that transductions in germline arise from a more diverse set of source L1s.

### Short transductions evade detection but are detectable with tag-based approaches

To assess the relative activity of source L1s across platforms and in a larger cohort of CRCs, we compared somatic insertion rates in our Nanopore-sequenced samples to those obtained from short-read whole-genome sequencing datasets. Short-read comparisons included an in-house CRC cohort (344 tumors with corresponding normals, where 50 overlapped with the Nanopore dataset) analyzed with xTea[29] (Methods: TE and transduction detection), as well as publicly available datasets processed with TraFiC-mem[9,22] (Fig. 2a). For consistency, we restricted the PCAWG dataset to colorectal tumors (Supplementary Data S1). Due to greater sequencing depth, we would expect short read sequencing to produce more somatic insertions per sample. Nevertheless, several individual source L1s showed higher activity in our Nanopore data. To understand this discrepancy, we examined the length distribution of transduced sequences and found that source L1s with unexpectedly high activity produced short transductions (Fig. 2b). Only elements where the majority of transduction sequence was longer than 100 bp (22q12.1-1, 14q23.1-1) were detected more frequently in

any of the short read-based studies (Fig. 2a, b). In total, 71% (248/349) of transductions identified by Nanopore were shorter than 100 bp (Fig. 2b and Supplementary Data S4). These short transductions were rarely detected in the short-read datasets, likely due to the difficulty of mapping short, low information content sequences adjacent to highly repetitive source L1s.

To investigate the detection of short transductions further, we analyzed 50 CRC samples with transductions detected by TraDetIONS using nanopore data and by xTea using Illumina data. Of 341 somatic 3′-transductions insertions detected by Nanopore, 75 (22%) were also classified as transductions by xTea, with 55 (73%) of those sharing the same assigned source L1 (Supplementary Data S2). Notably, 87% (48/55) of the transduced sequences were longer than 100 bp. In contrast, Nanopore identified 174 insertions as transductions that xTea had classified as solo-L1s: 96% (166/174) of them had a transduced sequence shorter than 100 bp. These results further support our observation that short-read methods are biased toward detecting longer transductions and often fail to resolve the source L1s of insertions with short transductions.

As source L1s are prone to produce insertions with highly distinct transduction lengths (Table 1), missing short transductions has biased our catalog of active L1 elements. According to our data, many elements with previous reported activity[9,22] display substantially more frequent transduction events (6p24.1-1, Xp22.2-1) (Fig. 2a). Further, we identified 9 source L1s with detected somatic transductions in our data that have no reported activity according to prior studies[4,5,9,16,22,25] (Supplementary Data S1). The source L1s have minor activity, ranging from 1 to 5 new insertions. The majority, 7/9 of the elements were present in the reference genome (GRCh38) and 6/9 were fixed in the population according to TraDetIONS[28]. Of the seven active reference elements five produced transductions exclusively shorter than 60 bp, demonstrating that overlooking short transductions has led to not identifying these L1s as active.

To assess whether these short transductions are present but undetected in short-read Illumina data, we developed a tag-based strategy that searched short Illumina reads for exact 30 bp sequence from 3′ regions of candidate source L1s, (Methods: Tag-Based Detection in Illumina). Applying this method to 356 CRC short-read genomes, we identified 851 somatic transductions. The majority of them (94%, 883/941) displayed a hallmark of bona

**Table 1 | Features of transductions by their source L1s**

| | Somatic insertions | Median length | Median L1 content (%) | 5′ inversion rate (%) | Orphan rate (%) | Mode unique sequence in bp (n = transductions) |
|---|---|---|---|---|---|---|
| 6p24.1-1 | 78 | 342 | 86 | 33 | 10 | 78 (18) |
| Xp22.2-1 | 72 | 492 | 91 | 28 | 3 | 37 (31) |
| 22q12.1-1 | 34 | 588 | 7 | 41 | 50 | 818 (7) |
| 7q21.3-2 | 24 | 296 | 83 | 21 | 8 | 61 (15) |
| 4p15.31-1 | 18 | 426 | 95 | 22 | 0 | 72 (7) |
| 8q24.21-1 | 13 | 457 | 92 | 23 | 8 | 85 (9) |
| 14q23.1-1 | 11 | 437 | 50 | 27 | 36 | 671 (9) |
| 6q25.3-1 | 11 | 618 | 22 | 46 | 46 | NA |

Eight most somatically active L1 elements are shown in the table. The table shows the number of somatic insertions per source L1, their median length, the median L1 content in insertion, 5′ inversion rate, the rate of orphan transductions in the insertions and mode length of unique sequence, where insertions that share the modal unique sequence length is in parenthesis. Mode length for 6q25.3-1 is not reported, as all the transduced sequences were of different lengths.

fide retrotransposition events (target site duplication, endonuclease cut site or a polyA-tail).

With tag-based detection, we again found Xp22.2-1, 22q12.1-1 and 6p24.1-1 to be among the most active elements, validating our findings in Nanopore data. When comparing the relative frequency of tag-based detection to Nanopore, we consistently detected more transduction in the Nanopore dataset (Supplementary Data S2) likely due to very specific targeting of the tag-based approach. Of the ten most active sources, only 12p13.32-1 showed more activity in tag-based detection. This could arise for instance, from these elements activating in a later stage and not being detected with the moderate coverage of Nanopore sequencing.

To validate the transductions experimentally, we performed PCR targeted sequencing to 20 nanopore and 20 tag-based detected transductions (Methods: PCR and sequencing; Supplementary Info: Targeted transduction validation). We were able to validate the presence of insertion in 18/20 Nanopore calls and 13/20 short read tag-based calls. The presence of transductions was validated for 18/18 and 8/11 insertions (Supplementary Data S5). Ten Nanopore calls were detected with xTea as well, however, they were detected to be solo-L1 insertions. Of these, 10/10 were validated to contain the transduction (Supplementary Data S5). The nanopore called transductions were shown to be true somatic as corresponding normal samples showed no band in similar conditions. One of the Illumina calls showed a faint band in the normal, possibly a result of somatic mosaicism or contamination (Supplementary Data S3). These findings support that our calls detect true transduction events although left undetected with previous detection.

Together, these results indicate that short 3′ transductions frequently accompany somatic L1 insertions but are systematically underdetected by current short-read sequencing analysis methods. This technical bias has previously skewed our view of the landscape of active L1 elements in cancer genomes, favoring L1s that generate longer, more easily resolved transductions.

### Insertion length variation reflects 5' inversion status and germline origin

Leveraging the advantages of long-read Nanopore sequencing, we investigated how L1 insertion length varies across biological and mechanistic contexts. We first assessed how insertion length relates to insertion class (solo, partnered, orphan). We found that partnered transductions are the longest (median: 445 bp), orphan transductions the shortest (median 335 bp) and solo-L1 (median 390 bp) fall in between ($p < 4 \times 10^{-4}$ Kruskal–Wallis rank sum test, Fig. 3a and Supplementary Data S6) and the same effect is seen in the germline (medians: 1520 bp, 968 bp, 1277 bp, respectively) ($p = 0.017$ Kruskal–Wallis, Fig. 3a), as reported[16]. Furthermore, germline insertions are longer in all insertion classes (Fig. 3a). To investigate whether this difference arises from distinct elements activating in the germline and in soma, we investigated the lengths within source L1s that

were activated in both (producing more than 2 somatic and germline insertions). The same pattern emerges within all source L1s (Fig. 3b), suggesting that germline insertions are longer than somatic insertions for reasons that cannot be explained by characteristics of individual source L1s.

When features of insertions are examined in context of their source L1s, we found that median insertion lengths differ between insertions from different source L1s ($p = 0.01556$, Kruskal–Wallis rank sum test, Table 1 and Supplementary Data S4). In addition to insertion lengths, the rate of 5' inversions varies between source L1s (Table 1 and Supplementary Data S4). To investigate the role the insertion sequence has in the formation of inversions, we analyzed the loci of inversion-related breakpoints within the L1 sequence (Methods: Inversion analysis). L1 elements themselves are prone to inversions in multiple loci, most prominently 250 bp upstream of their 3' end (Supplementary Fig. S4a). This supports the finding that insertions with L1 sequence, solo-L1s and partnered transductions, have higher frequency of inversions (30%, 30%) than orphan transductions with no L1 sequence (16%) (Supplementary Data S7). Additionally, when insertions are divided based on their source L1s, only sources with a high inversion rate contain breakpoint clusters in their downstream transduced sequence (Supplementary Fig. S4b), showing that source L1s with high inversion rate contain additional inversion sites beyond shared L1 sequence.

As previously reported[28], insertions with a 5′ inversion event tend to be longer than insertions without an inversion. When comparing the inversion rates for source L1s and their median lengths, a trend emerges: The higher the 5' inversion rate the longer the median insertion length (Table 1). The same effect can be seen with insertion types: rate of 5′ inversions follows the median lengths of insertion classes both in germline and somatic insertions (Supplementary Data S7). Thus, when the presence of 5′ inversions are taken into account, the insertion length differences between insertion classes and source L1s becomes less pronounced (Fig. 3c). In addition, the difference in insertion lengths per source L1 becomes nonsignificant when the effect of 5' inversions are taken into account (Kruskal–Wallis rank sum test, $p$ values 0.09871 and 0.3016; Supplementary Data S4). This suggests that two major components factor into insertion length: the presence of 5′ inversion and the somatic status of the insertion.

### Transductions are often terminated by polyadenylation signals

The length of transduction varies between different source L1s and is often limited to distinct values (Fig. 2b and Table 1). To investigate factors contributing to this variation in transduction length, we assessed the role of polyadenylation signal (PAS) strength downstream of active source L1s. The loci and strength of PAS is expected to influence transduction length, with strong signals near the 3' end of source L1s producing short transductions.

We applied a position weight matrix model[30] to quantify PAS strength downstream of each source L1 (Methods: Polyadenylation detection; Supplementary Data S8). Canonical PAS motifs typically reside 10–30 bp upstream of transcription termination sites[11], and accordingly, for 274 of

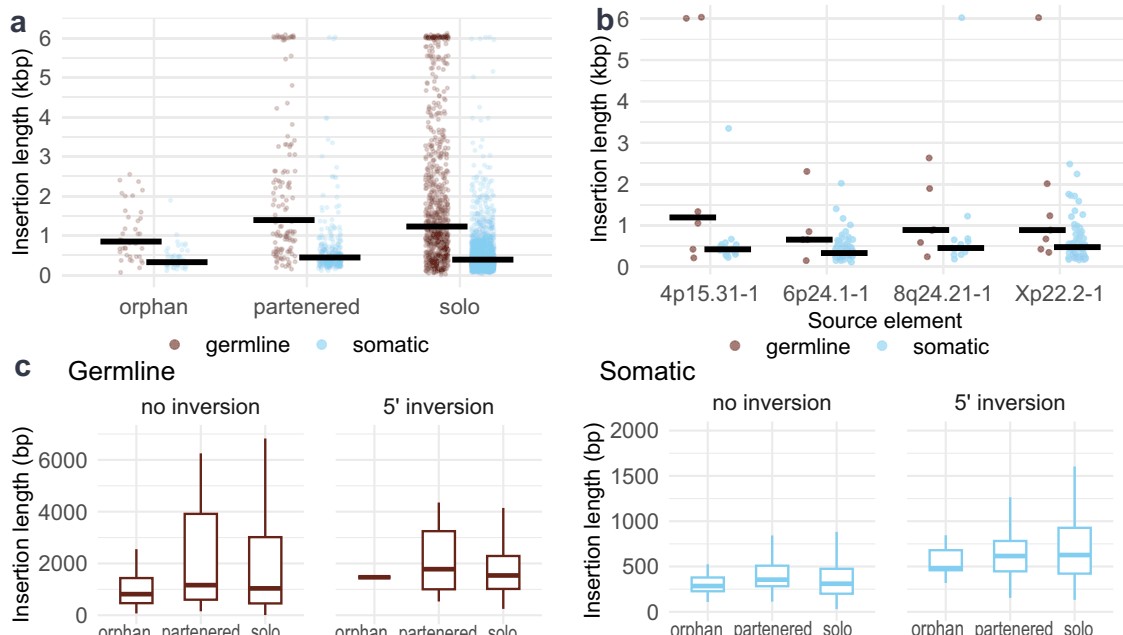

**Fig. 3 | Insertion lengths between insertion classes, tissue types and 5′ inversion status. a** Strip plot of insertion lengths in germline and in soma divided to different insertion classes. Insertion length is the length of the sequence inserted into the target consisting of L1 sequence, transduction and of polyA-tail. The median length of the group is shown as a black bar. **b** Insertion lengths in germline and in soma divided based on their source L1s. The median length of the group is shown as a black bar. **c** Boxplots of insertion lengths divided by their insertion class, the presence of 5′-inversion and the somatic/germline division (sample sizes for germline $n = 33$, 93, 552, 1, 26, 229 and for somatic $n = 13$, 95, 434, 39, 202, 1033 respectively).

349 (79%) transduction endpoints a PAS (score >8.1) occurred within this window. As an example, the canonical PAS 9 bp downstream of Xp22.2-1 terminated 68/72 transductions arising from the L1 (Fig. 4a). However, not all frequent termination points coincided with a strong PAS, and we identified two abnormal patterns associated with transduction termination. First, several recurrent endpoints in 6p24.1-1 lacked a high-scoring PAS motif near the insertion boundary (Fig. 4a), suggesting transcriptional termination can occur in the absence of a strong PAS. Second, with source L1 22q12.1-1, transcription often bypasses the internal canonical PAS and terminates further downstream at a lower-scoring site (Fig. 4a and Supplementary Fig. S5), indicating that canonical PAS do not necessarily act as effective termination signals. Further examples of these patterns exist for other source L1s as well (Supplementary Fig. S5).

To understand if the strength of the PAS inside source L1 affects whether L1s produce transductions, we calculated the strength of PAS inside reference L1HS elements. Transductionally active elements were enriched for both weak and strong PAS (Fig. 4b). While inactive elements contained mainly the canonical PAS (84%, 247/294), 4% of inactive elements (12/294) harbored only a weak PAS (score <8), suggesting a potential predisposition to generate transductions if transcriptionally active.

These findings indicate that PAS architecture contributes to transduction length and termination site choice[31], however, transcriptional termination is not solely determined by PAS strength and may involve additional sequence or structural features.

**Atypical and multigenerational transduction events complicate the transduction landscape**

Nanopore sequencing provides access to internal sequences of the insertions. By creating detailed annotations of them, we were able to identify unusual cases of transduction events. We present four source L1s, with unique features that showcase the complexity of transductions and complicate their detection.

Firstly, we identified a multigenerational L1 transduction cascade: a reference L1 at chrX:1193529–1194131 giving rise to a germline insertion at chr15:84597670, carrying identifiable 3′ transduced sequence. This germline

insertion was subsequently mobilized in the soma, generating a somatic insertion at chr11:2170882 that retained the original 3′ tag from the reference L1, as well as a second transduction tag from the intermediate germline insertion.

The second unusual transduction event is a somatic full length insertion 6p12.3-1. Full length somatic L1s (>5990 bp) are rare events, as we report 5 of them, constituting 0.3% of all somatic L1s (5/1531). One of them, the insertion in 6p12.3-1 gave rise to five downstream somatic insertions in the corresponding tumor, being responsible for 5/68 of the somatic insertions in the sample. Along with previously reported examples[9,22], this demonstrates how novel full-length elements have insertional capabilities that can potentially have insertional cascading effect in the tumor.

Thirdly, we identified four somatic insertions derived from the L1 element in 9q31.3-1, all terminated at an identical genomic location—chr9:108,805,090 (GRCh38), corresponding to a PAS in a truncated reference L1 element situated ~2.4 kbp downstream of the source L1. Although these insertions included L1 sequence from the truncated downstream element, they did not include any sequence from the original 9q31.3-1 source L1 and thus were classified as orphan transductions.

The fourth case identified is a group of L1s we call the *proxy* elements. They are truncated L1 elements in the reference genome that contain transduced sequences from their source which itself is absent from the reference. Thus, the new offspring from the source L1 will be linked to the truncated element, as the proxy contains L1 sequence in addition to the transduced sequence for the alignment. We identified two proxy elements in the reference GRCh38 (Supplement: Reference proxies for polymorphic L1 elements) that were initially associated with transductions detected by nanopore sequencing. However, identification of proxy elements allowed correct reassignment of these transductions to their true sources. In total 83 somatic insertions were reassigned due to proxies.

Elements such as the L1 9q31.3-1 and the proxy elements are examples of the complex nature of retrotransposition challenging the identification of active sources. They are linked to truncated elements rather than their true sources and thus could be dismissed. Their presence highlights the benefits

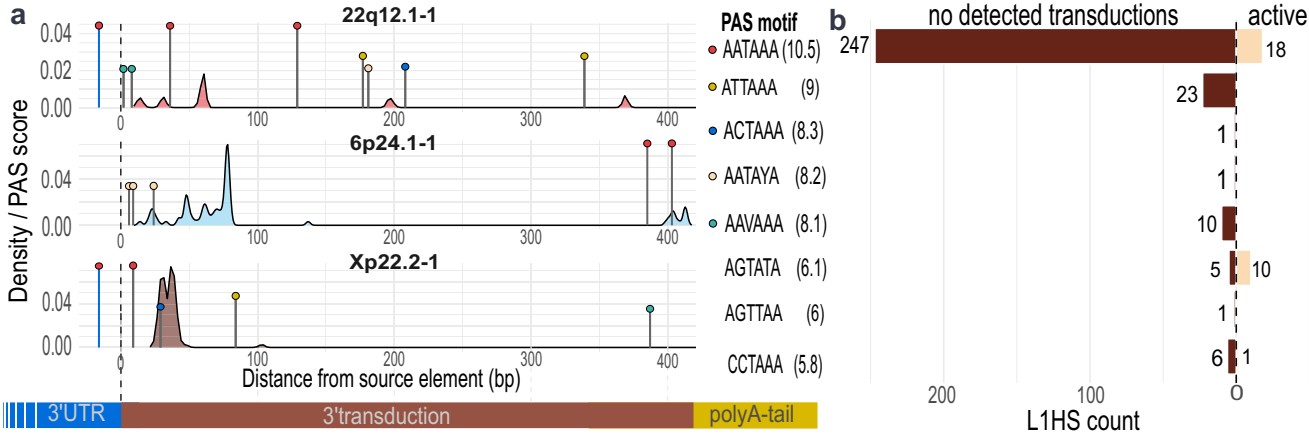

**Fig. 4 | Polyadenylation signals downstream and inside source L1s. a** Density plot of transduction end points downstream the most frequent source L1s limited to 400 bp downstream of the L1 sequence. Lollipops present the PAS with score >8.1, where their height corresponds to the signal strength and their color to the PAS sequence. The two reference elements have their internal PAS represented by a blue lollipop at the start of the plots. **b** The frequency of the internal PAS within 50 bp of the reference L1HS elements 3' end. Active elements have detected somatic transductions, whereas others have not.

of long read sequencing that provide the opportunity to assign more complex insertions to their source L1s.

## L1 promoter hypomethylation is associated with transduction activity across loci and tissues

L1 promoters are typically methylated in normal cells[32], but they are known to experience hypomethylation in various cancers, including CRC[33,34]. Furthermore, the source L1 promoter methylation has been associated with their activity in cell lines[16,20], prompting us to investigate whether L1 promoter methylation correlates with somatic L1 transduction activity in CRC. To address this, we used nanopore long-read sequencing to quantify promoter methylation levels of reference source L1s in CRC tumors and normal tissues.

We characterized the methylation patterns of different source L1 elements using nanopore sequencing (Methods: Analysis of DNA methylation). Visualization of the methylation patterns around two active reference L1 elements (22q12.1-1, Xp22.2-1) revealed a clear decrease in methylation values at the area preceding and overlapping the L1 element in both CRC tumors and in normal colon (Fig. 5a, b and Supplementary Fig. S6) in concordance with prior literature[16]. The most significant decrease was observed in CRC tumor samples, especially in samples showing transduction activity (Fig. 5a, b). The X-linked element Xp22.2-1 also revealed sex-specific differences: male CRC samples with transductions showed a robust reduction in promoter methylation. On the other hand, women harbor two copies of X-chromosomes, yet due to dosage compensation, one chromosome copy is silenced during early development. The methylation signal in Xp22.2-1 was diminished in females, likely decreased only in the active X-chromosome copy, weakening the signal (Supplementary Fig. S7).

To generalize the connection between L1 methylation and transduction activity we calculated the average methylation values for 100 bp up- and downstream sequences around L1HS 5' ends. As a case study, we used source L1 with activity in most samples, 22q12.1-1, and compared the number of transductions in the samples to this average methylation of the source L1. Samples with transductions exhibited lower average methylation than those without, although variation was present among transduction-negative samples (Fig. 5c) ($r = -0.274$, 95% CI [$-0.501$, $-0.012$], $p = 0.041$). Expanding this analysis to all fixed source L1s with sufficient coverage (≥75% of CRC samples; $n = 14$), we found that methylation levels were significantly lower in samples with detectable transduction activity compared to those without ($p = 0.0044$, 95% CI [$-16.5$, $-3.23$], Welch's $t$-test; Fig. 5d). These results support a link between local promoter hypomethylation and source L1 activation.

As source element methylation had been shown to correlate with transductional activity in CRC, we augmented the analysis to include other tissues as control sets: normal colon, uterine leiomyoma (UL) and myometrium. We classified the source L1s based on their observed activity in CRC: active (>3 transduction-positive CRC samples), low activity (1–2 samples), or inactive (no somatic transductions). CRC samples depicted the lowest L1 methylation levels when compared to other tissue types. The average methylation values for L1 elements with transductions were lower compared to the methylation values collected from inactive elements (Fig. 5e). Unexpectedly, methylation differences between active and inactive L1 elements were not observed only in CRC but were also present in all tissue types studied. However, differences in other tissues were not as drastic as in CRC.

Finally, we examined a subset of inactive L1 elements with low PAS scores (<8, $n = 7$) (Fig. 4b). Low PAS score indicates a strong transduction potential, yet according to our analysis, these elements have remained silent in our data, suggesting that these elements are truly inactive. As expected, they depicted higher average methylation levels than other inactive L1 elements (Fig. 5e and Supplementary Fig. S8). This result reinforces the link between promoter methylation and source inactivity.

## Discussion

In this study, we characterized the activity of source L1 elements in colorectal cancer. We approached the target by leveraging long read sequencing of 56 CRC samples and creating precise recreations of the inserted sequences. By mapping the insertion sequences back to their genomic origins, we were able to identify the active source elements. Using resolved insertion sequences provided a high-resolution view of active L1s contributing to the insertional burden of the tumors.

We detected source L1s by utilizing 3' transductions, leaving a large portion of somatic activity unassigned. The remaining sources are challenging to identify, as L1 sequences are largely shared among different loci, with only occasional internal polymorphisms. Leveraging such polymorphisms for source detection would require substantially higher sequencing depth, as nanopore sequencing remains characterized by a relatively high error rate.

We found that the majority of 3' transductions originated from a small subset of highly active L1 elements. Remarkably, just three elements accounted for 53% of all somatic transductions, consistent with prior reports that a limited number of hot L1s accounts for most of transductions in cancer[4,9]. Moreover, the number of somatic transductions and solo-L1 in

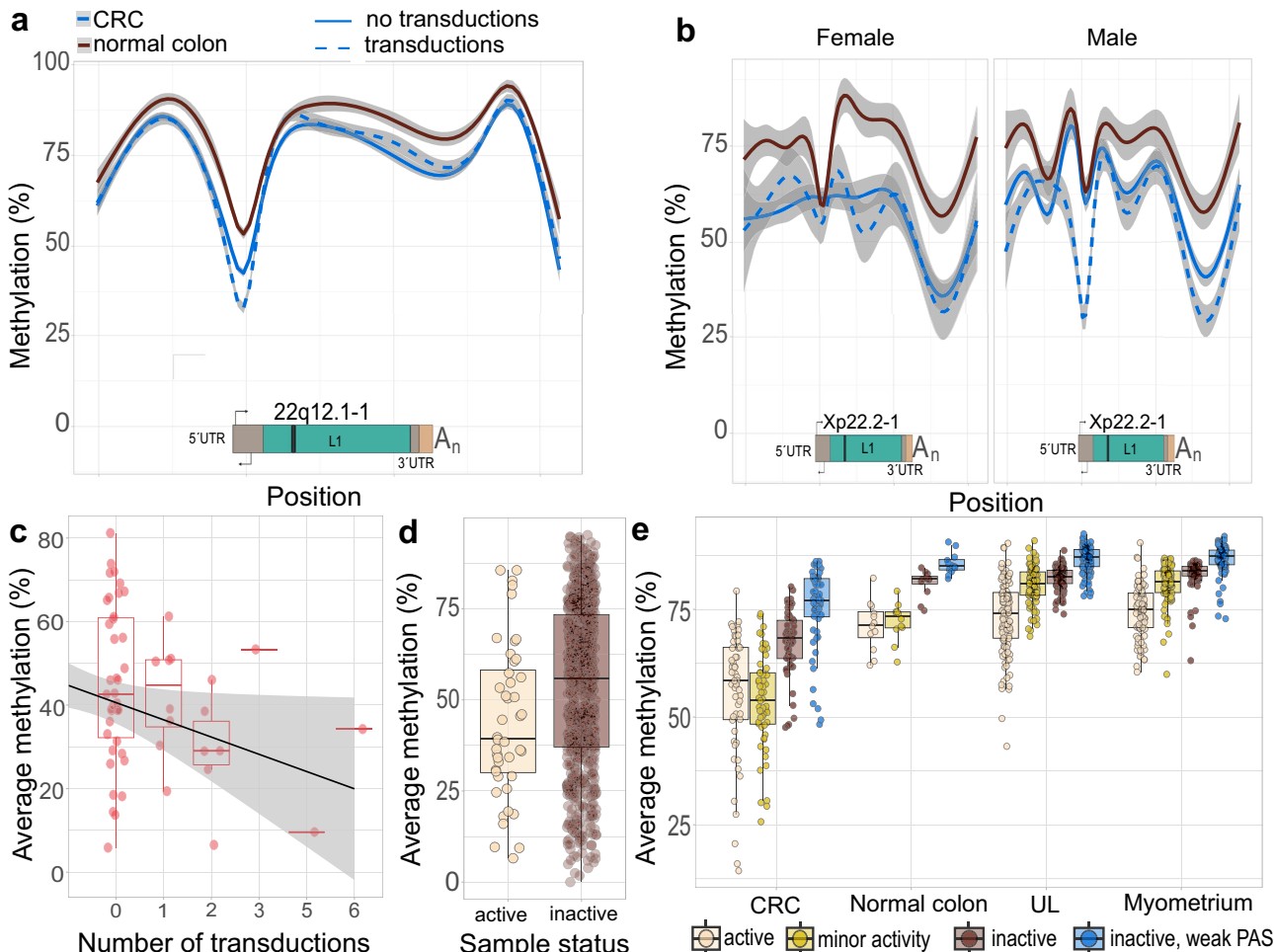

**Fig. 5 | Promoter methylation of active and inactive source L1s. a** Smoothed methylation curves around 22q12.1-1 from 12 normal colon samples and 56 CRC samples: methylation curves for CRC samples with transduction activity ($n = 17$, dotted line) and samples without detected transduction activity ($n = 39$, solid line) are shown separately. **b** Smoothed methylation curves for Xp22.2-1 are shown separately for females (transduction activity, $n = 2$; no transductions, $n = 7$) and males (transduction activity, $n = 8$; no transductions, $n = 23$). Xp22.2-1 is polymorphic: data is shown only for homozygous (2 copies in females, 1 copy in males) individuals. **c** Average methylation at a 200 bp region around source L1 22q12.1-1 5' area vs. the number of transductions arising from it. Each point is one CRC sample

(sample sizes: $n = 38, 8, 6, 1, 1, 1$). **d** Average methylation values from 14 fixed reference L1 elements with activity detected in our data. Each CRC sample contributes one point for each of the 14 L1 elements (active $n = 44$, inactive $n = 730$). Samples with transduction activity have lower L1 methylation values. **e** Average methylation levels for L1 elements based on their activity status. Each point depicts average methylation from multiple L1 elements of a certain activity type in one sample. Average methylation values are shown for CRC ($n = 56$), normal colon ($n = 12$) and for comparison purposes, for uterine leiomyomas (ULs) (106) and normal myometrium (92).

samples showed a strong positive correlation, which enables the use of transductions as indicators for overall activity.

Most transduced sequences in our dataset were relatively short, typically containing less than 100 bp of unique downstream sequence. Short transduction sequences, often flanked by polyA-tails on both sides, have been hard to detect with short read sequencing. Utilizing samples sequenced with both short and long read sequencing demonstrated that short transductions have been detected as solo-L1 insertions, with their sources left undetected. To further validate these findings, we applied a targeted transduction detection method to a large set of short read CRC samples and confirmed that highly active source L1s producing short transductions, such as Xp22.2-1 and 6p24.1-1, were also active in the larger set of samples. Furthermore, targeted PCR experiments validated the presence of several of the transductions, reinforcing the accuracy of our long-read-based approach.

As source L1s have variable transduction lengths, the sources prone to produce longer transductions, such as 22q12.1, have been overrepresented in previous short read studies. While many active sources mainly producing short transductions had been identified in them, the extent of the somatic

insertions had been underestimated[9,22,25]. Additionally, our long-read approach revealed a broader landscape of source activity. We identified 9 previously unreported somatically active source L1s, several of which exclusively generated short transductions, highlighting the value of long-read sequencing in detecting source activity.

We introduced a class of truncated reference elements, which we termed *proxy* elements. They contain transduced sequences derived from their source L1 absent from the reference. As a result, insertions originating from the true source L1 were often incorrectly attributed to these proxy sites. In our dataset, two proxy elements accounted for a total of 83 initially misassigned somatic insertions, further highlighting the importance of long-read sequencing approaches when studying insertional activity in cancer.

Consistent with previous reports[16], insertion lengths followed a decreasing trend from partnered transductions to solo-L1s to orphan transductions. The differences between the insertion types disappeared when taking into account their varying 5' inversion rate. The effects of insertion class and 5' inversions were present in germline L1s as well: the insertion length followed the inversion rate between insertion classes. Source L1s also exhibited the effect of 5'inversions: different sources

produced insertion of varying lengths but, however, the difference disappeared when taking the varying 5' inversion rate of source elements into account.

As the unique transduction sequences often are of highly distinct lengths based on their source L1s, we scrutinized the polyadenylation signals (PAS) downstream of the sources. As expected, strong PAS signals were found 10–30 bp downstream of frequent transduction end sites, however, this did not apply to all of them[12]. Further exceptions arose when calculating the scores of PAS inside the elements themselves: often transcription bypassed the canonical and the strongest PAS and instead terminated on further downstream PAS with a lower score consistent with a previous study[31]. In addition, many of the active source L1s produce transductions despite containing the canonical strongest PAS. These findings suggest that while the PAS and their strength play a role in the genesis of transductions and their termination, other factors are likely to contribute.

Nanopore sequencing provided access not only to the internal L1 sequences but also to information on DNA methylation. Since multiple studies have reported increased L1 activity associated with promoter hypomethylation[9,26,35,36], we examined the methylation levels of active source elements. Analysis of methylation levels in the vicinity of active source L1s revealed a clear decrease near to 5' regions, especially in CRC samples with transduction activity from the corresponding L1 elements. This pattern of methylation decreases in the 5' regions has also been observed in normal colon with detected L1 activity[16]. Transductional activity of L1 elements was also associated with methylation levels on a broad scale, as active elements showed on average lower methylation levels than elements without transduction activity. Unexpectedly, a similar, albeit a less pronounced pattern emerged in normal colon, uterine leiomyoma (UL) and myometrium: L1s displaying somatic activity in CRC depicted lower methylation in contrast to inactive L1s in other tissues as well. Normal colon has been shown to experience minor somatic L1 activity[16], but despite scrutiny, no L1s have been detected to create somatic transposons in UL[28]. Low L1 methylation levels in UL samples, however, do not indicate insertional activity but likely reflect an association with genomic locations showing low methylation values in general. L1s and their surroundings have shown interaction with methylation[20], which could indicate that many active elements reside in lowly methylated regions. However, due to cancer associated hypomethylation, the methylation levels of L1s decrease even further in CRC, potentially allowing activation of L1s in CRC.

The highest methylation levels of source L1s were among the inactive sources with a weak internal PAS. Due to weak PAS, activity in them would likely lead to identifiable transductions. Thus, the absence of transductions indicates high confidence inactivity. The high methylation of these elements showcases that methylation is contributing to their silence. Additionally, the inactive sources with strong PAS with lower methylation levels could indicate that a subset of them are active but producing unidentifiable solo-insertions.

Our results demonstrate how methylation levels are associated with transduction activity on two separate levels: firstly, CRC samples with somatic transductions have lower methylation in the active sources compared to samples without activity. Secondly, L1 elements showing transduction activity have in general lower methylation levels compared to inactive L1 elements. However, we also observed inactive L1 elements that harbored low methylation levels. Furthermore, source L1s expressing activity in other samples, occasionally depicted low methylation also in samples without activity. This could be attributed to these L1s having a strong PAS and producing solo-L1s, as ~75% of somatic insertions do not contain sequence outside L1. Thus, L1s with low methylation but no transductions may just be prone to produce solo-L1s. Moreover, sequencing coverage limits our means to detect subclonal insertions, indicating that L1 activity in certain samples may be present yet remain undetectable without higher sequencing coverage. Furthermore, the detected somatic L1 events are a cumulation of somatic events during tumorigenesis that may not correspond to methylation levels in the moment of sequencing[37].

Additionally, DNA methylation is not the only factor contributing to the silencing of L1s[38,39].

Together, our findings provide a comprehensive map of somatic L1 retrotransposition in colorectal cancer, uncovering a large number of short transductions which had most likely been detected as solo-L1s in earlier studies. By leveraging long-read sequencing, we resolved the structure and origin of insertions with unprecedented precision and integrated this information into methylation of the L1 elements. While low promoter methylation correlates with transductional activity, our results in other tissues suggest that methylation alone may not be sufficient to predict activation. This work highlights the importance of integrating sequence- and epigenome-level information to understand the dynamics of L1 mobilization in cancer.

## Methods
### Sample collection
The study was reviewed and approved by the Finnish National Supervisory Authority for Welfare and Health, National Institute for Health and Welfare (THL/151/5.05.00/2017, THL/723/5.05.00/2018, THL/1300/5.05.00/2019), and the Ethics Committee of the Hospital District of Helsinki and Uusimaa (Dnro 133/E8/03, 408/13/03/03/2009, HUS/2509/2016, 177/13/03/03/2016). Informed consent was obtained from all human participants. All ethical regulations relevant to human research participants were followed.

This study used whole genome sequenced tumor samples to investigate the activity of source L1 elements in colorectal cancer. We utilized 359 colorectal cancer (CRC) and 358 corresponding normal colon tissues, along with 132 uterine leiomyomas (UL) and 107 matched myometrium samples as a control group. The CRC samples and their adjacent normal tissues were fresh-frozen specimens obtained from a population-based cohort of Finnish CRC cases[40,41]. Similarly, the UL samples, consisting, were of fresh-frozen tumors harvested post-hysterectomy as previously described[42]. Pathological review confirmed that all tumor samples included in the study contained at least 50% tumor content.

### DNA extraction
Genomic DNA was extracted from both tumor and matched normal tissues of CRC patients using standard DNA isolation protocols[43]. For uterine leiomyomas (UL) and their corresponding myometrium samples, DNA was isolated using the QIAamp Fast DNA Tissue Kit (Qiagen).

### Sequencing
**Illumina sequencing**. Illumina sequencing sample set included 356 CRC samples, 26 UL samples, 356 colon, and 26 myometrium corresponding normal tissues. Illumina sequencing was conducted following established protocols[44,45].

**Nanopore sequencing**. Nanopore sequencing utilized 56 CRC samples with 12 matched normal colon, along with 106 UL samples and 96 matched myometrium tissues. Sequencing was performed using Nanopore technology as previously described[28].

### TE and transduction detection
**Source TE selection**. To annotate transductions, we utilized a curated set of source L1 elements. These included full-length L1 elements (5700–6700 bp) annotated in the GRCh38 reference genome[46] using the Repeat Browser[47], as well as germline and somatic full-length L1 elements identified from nanopore sequenced samples[28]. In addition, the elements were supplemented with reported active source L1s from literature[9,22].

**Transduction detection in nanopore**. We utilized long read sequencing to detect transduction and solo-TE insertions from tumor and corresponding normals. Transductions and solo-TEs were detected from Nanopore sequenced samples using SVs called by Sniffles[48] as previously described[28]. Transductions were identified by the presence of sequence mapping to the reference genome within 3 kbp of the 3' end of potential

source L1 elements, with MAPQ > 0. Insertions containing L1 sequences were classified as partnered transductions, whereas insertions lacking the sequence were classified as orphan transductions. The detection was performed as previously reported[28], however, we incorporated additional source L1s (source TE selection). Although the majority of the CRC tumors lacked a nanopore sequenced corresponding normal, we utilized the use of all the other samples as a control. This resulted in somatic L1 calls with 93–96% rate of true somatic events as evaluated by visualization and discordant read counts with short read WGS corresponding normals and targeted PCR experiments[28].

**Tag-based detection in Illumina**. The transductions were detected in short read WGS. Tag-based detection relied on the presence of an exact 30 bp sequence located downstream of known full-length L1 elements. To reduce false positives due to repetitive sequences, we quantified the occurrence of tag sequences in the human genome using edlib[49], filtering out sequences present in more than 1000 locations, allowing for up to three insertions, deletions or substitutions. This resulted in the use of 338 elements as the potential source L1s. In addition, we recorded the positions of similar sequences in the reference allowing up to 6 mismatches with edlib to use in further filtering.

For each potential source L1, we identified read pairs containing these tag sequences. If one read of the pair had a MAPQ ≥ 37, it was considered an anchor read. Anchor reads were merged in a cluster if within 1 kbb of each other. If a cluster of reads contained ≥3 reads and the cluster was not within 2 kbp of a similar sequence in the reference genome, it was considered a transduction. If transductions were within 2 kbp of transductions in other samples, arising from the same sources, they were considered as the same insertion. Insertions present in only one tumor were considered somatic.

Each transduction identified through the tag-based approach underwent additional characterization, consisting of breakpoint detection, retrotransposition hallmark identification, and L1 sequence identification. We extracted reads within 500 bp of the anchor reads and identified split reads and discordant reads (paired-end distance >2 kbp, pairs mapped to different chromosomes or incorrect orientations). The insertion breakpoints were detected from split reads with the most common split positions assigned as breakpoints. Target site duplications were defined as the distance between breakpoints. Poly-A tails were detected in sequences of discordant reads by using sweep line technique (+1 for match, −3 for mismatch and 10 as the limit). EN cut site detection was performed by analyzing breakpoint flanking sequences from the reference genome. The polyA-tail marked the orientation of the insertion (tract of As +, Ts −). Sequence surrounding the initial breakpoint (former if − orientation, latter if + orientation) was extracted asymmetrically containing four bases upstream and two bases downstream oriented insertions. The sequence was scored based on the similarity to the consensus sequence TTTT/AA: every base in the flanking sequence that matched with base and position to consensus, received a point, except 3rd and 4th base, where the points were 3 and 4 respectively. Sequences with the final score ≥6 were considered as EN cut sites.

**xTea detection in short read data**. To perform a short read-based detection as a control, we used xTea[29] to detect TE insertions from short read sequenced samples. We used mainly the same samples as with the tag-based methods, however, 12 xTea runs failed to produce output, thus leading the sample set to consist of 344 CRC samples and 26 ULs 26 normals. Default parameters were used in the running of xTea.

**PCR and sequencing**
To validate the sequence contents and the somatic nature of the detected transductions, we validated somatic transductions with PCR and Sanger sequencing. We validated in total 40 different somatic insertions arising from 22 samples. For both nanopore and tag-based detection, this consisted of 10 random somatic insertions and another 10 identified as transductions but called as solo-L1 elements by xTea. Primers were designed so that the other primer was outside the insertion (designed with Primer3, http://

primer3.ut.ee) and the other spanning the insertion site or containing inserted L1 sequence and primers spanning or inside the insertion (hand-designed) (Supplementary Fig. S1). PCR was run in the same conditions for tumor and corresponding normal. We performed the PCR and Sanger sequencing as previously described[50], except that PCR purification was performed with A'SAP PCR clean-up kit, cat no 80350-2000, ArticZymes Technologies.

As Sanger sequencing did not enable us to sequence the transduction sequence flanked by two polyA-tails (Supplementary Fig. S1), we sequenced the PCR products also with Nanopore sequencing. Nanopore amplicon sequencing was performed following the manufacturer's protocol for SQK-LSK114 amplicon sequencing. Reads were assembled and corrected with 'hifiasm --ont'[51] and mapped to GRCh38 reference with minimap2[52]. A transduction was called validated if a raw unitig or a corrected read had supplementary alignments on both the target and source genomic regions.

Additionally, to detect transduction sequence, we designed PCR primers that paired directly to transduction (Supplementary Fig. S1). We performed Sanger sequencing as before.

**Inversion analysis**
5′-inversions were detected from insertions based on mapping to L1 sequence or reference in two orientations[28]. The two inversion breakpoints occurring in one event were clustered together with all inversion breakpoints in the context of their distance to 3' end of L1. Rectangular kernel with bandwidth 10 was used in the clustering.

**Polyadenylation detection**
We detected polyadenylation signals (PAS) by applying a position weight matrix model[30] and quantifying PAS strength downstream of active source L1s. We applied the detection to downstream reference sequences of the elements extending to 50 bp over the furthest transduction end site. Next, we looked for the internal PAS in the final 50 bp inside reference L1HS elements[47]. If multiple were found, we selected PAS with the strongest score.

**Analysis of DNA methylation**
DNA methylation calling from nanopore long-read sequencing data was performed using guppy/megalodon 2.4.2. Reads were aligned to the T2T-CHM13v2.0 reference with minimap2[52]. Phasing to haplotypes was performed with Longshot v.0.4.3[53].

R-package ggplot2 was utilized in the visualizations[54]. Methylation curves around 22q12.1-1 and Xp22.2-1 were drawn with the function geom_smooth() using method='gam'. Average DNA methylation levels for 100 bp inside and outside (200 bp in total) of the L1 5' areas were defined as the average methylation of the CpGs overlapping this region. CpGs with a coverage of 3-75 reads were included in the analysis. L1 elements were divided into 4 categories based on their nanopore sequencing based transduction activity in our CRC collection: (1) Active L1 elements, referring to L1 elements with activity detected in at least 4 CRC samples; (2) minor activity L1 elements, referring to L1 elements with transductions detected in 1–3 CRC samples; (3) Inactive L1 elements, referring to L1 elements without transductions; and (4) Inactive L1 elements with a weak PAS. An element has been included in the average methylation analyses if (a) the 200 bp region contained at least 3CpGs and (b) at least 75% of the samples analyzed had methylation data available. Analyses utilizing average methylation levels were limited to autosomal reference L1 elements with fixed genotypes in the population.

**Statistics and reproducibility**
Statistical analyses were conducted using R and Python with specific tests detailed in Results. Sample size was not predetermined by statistical considerations; the study utilized all available samples meeting quality criteria. Experiments were not randomized or blinded.

The study used three sequencing platforms and detection methods: Oxford Nanopore sequencing of 56 CRC and 12 normal colon samples analyzed with TraDetIONS, Illumina sequencing of 356 CRC and 356

matching normal samples (50 CRC samples overlapping with Nanopore dataset), analyzed with tag-based detection and xTea was applied to matching pairs. Validation included PCR amplification and Sanger/nanopore sequencing of 40 somatic insertions in 22 tumors

## Reporting summary

Further information on research design is available in the Nature Portfolio Reporting Summary linked to this article.

## Data availability

The colorectal cancer dataset analyzed during the current study is not publicly available due to concerns regarding patient anonymity and lack of consent for genomic data sharing for samples collected decades ago, as required by Finnish law and the EU General Data Protection Regulation (GDPR). The data are available from the corresponding author on reasonable request and on a collaborative basis. The UL samples can be obtained via FEGA. Large additional datasets are available in Zenodo https://doi.org/10.5281/zenodo.17293681. They include somatic insertion sequences along with their annotations. Additionally, it contains 5' methylation of reference L1-HS in our samples.

## Code availability

The pipeline TradetIONS used for transduction detection in Nanopore data is available on GitHub https://github.com/panummi/TraDetIONS. Pipeline used for detecting tag-based transductions in Illumina data is also available https://github.com/panummi/Tag_based_transduction_search. Both are also available in Zenodo (https://zenodo.org/records/17924887, https://zenodo.org/records/13284044) as a record of archival[54–56].

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

## Acknowledgements

We thank S. Marttinen, S. Soisalo, M. Rajalaakso, I.-L. Åberg, I. Vuoristo, A. London, H. Metsola, K. Pylvänäinen, B. Björkroth and P. Ikonen for technical support. We thank A. London for proofreading the manuscript. We acknowledge the computational resources provided by the ELIXIR node, hosted at the CSC–IT Center for Science, Finland. This work was supported by grants from Research council of Finland (Finnish Center of Excellence Program 2018–2025 [No. 352814]; Academy Professor grants [No. 319083, 320149]; iCAN Digital Precision Cancer Medicine Flagship [320185]); Cancer Foundation Finland [230068, 200071]; Sigrid Juselius Foundation [230002]; Jane and Aatos Erkko Foundation [220001]; The Emil Aaltonen Foundation; Biomedicum Helsinki Foundation; Orion Research Foundation sr; Juhani Aho Foundation for Medical Research; The Paulo Foundation; and The Ida Montin Foundation.

## Author contributions

P.N. performed method development, data analysis, and study design; A.T. performed methylation analysis and study design; J.R., T.N., and N.V. performed data analysis; A.L., L.R.-S., S.K., T.T.S., A.R., K.T., A.M., J.B., J.-P.M., E.S., A.P., O.H., and R.B. collected the tissue samples; A.K. performed validation design; K.P. performed data analysis, study design and supervision; T.C. and L.A.A. performed study design and supervision. All authors were involved in writing and approved the final version of the manuscript.

## Competing interests

The authors declare the following competing interests: T.T.S. reports consultation fees from Mehiläinen, Nouscom, Orion Pharma, Amgen, and Tillots Pharma, and a position in the Clinical Advisory Board and as a minor shareholder of Lynsight Ltd. The other authors declare no competing interests.

## Additional information

[1]Applied Tumor Genomics Research Program, Research Programs Unit, University of Helsinki, Helsinki, Finland. [2]Department of Medical and Clinical Genetics, Medicum, University of Helsinki, Helsinki, Finland. [3]Molecular and Integrative Biosciences Research Programme, Faculty of Biological and Environmental Sciences, University of Helsinki, Helsinki, Finland. [4]Department of Computer Science, University of Helsinki, Helsinki, Finland. [5]iCAN Digital Precision Cancer Medicine Flagship, University of Helsinki, Helsinki, Finland. [6]Department of Gastrointestinal Surgery, Helsinki University Central Hospital, University of Helsinki, Helsinki, Finland. [7]Faculty of Medicine and Health Technology, University of Tampere and TAYS Cancer Centre, Tampere, Finland. [8]Department of Gastroenterology and Alimentary Tract Surgery, Tampere University Hospital, Tampere, Finland. [9]Abdominal Center, Helsinki University Hospital, Helsinki University, Helsinki, Finland. [10]Department of Pathology, HUS Diagnostic Center, Helsinki University Hospital and University of Helsinki, Helsinki, Finland. [11]Department of Surgery, Wellbeing Services County of Central Finland/ Hospital Nova of Central Finland, Jyväskylä, Finland. [12]Department of Education and Research, Well Being Services County of Central Finland, Jyväskylä, Finland. [13]Department of Health Sciences, Faculty of Sport and Health Sciences, University of Jyväskylä, Jyväskylä, Finland. [14]Department of Obstetrics and Gynecology, University of Helsinki and Helsinki University Hospital, Helsinki, Finland. [15]Department of Pathology, Dana-Farber Cancer Institute and Harvard Medical School, Boston, MA, USA. [16]Institute for Molecular Medicine Finland (FIMM), University of Helsinki, Helsinki, Finland. ✉e-mail: kimmo.palin@helsinki.fi; tatiana.cajuso@helsinki.fi

