## [Transparent Peer Review file · Communications Biology]

Nanopore sequencing reveals hidden landscape of short L1 transductions in colorectal cancer

Corresponding Author: Dr Kimmo Palin

Version 0:

Reviewer comments:

Reviewer #1

(Remarks to the Author)

This is a well-written manuscript that leverages one of the key advantages of nanopore sequencing to study L1 transductions in colorectal cancer. The data and conclusions are strong and contribute meaningfully to our understanding of retrotransposition, despite some minor limitations. Nummi et al. convincingly demonstrate that additional sequences generated through L1 transductions have often been missed due to technological constraints. By applying nanopore sequencing to colorectal cancer as a model system, they show that transductions occur more frequently than previously appreciated.

My comments:

1. It would be helpful to include a discussion of the limitations of nanopore sequencing in this context.
 2. The current framework for defining the landscape of "active" L1s relies entirely on transductions, which represent only ~23% of insertion events. This approach introduces a systematic bias by potentially overlooking highly active elements that generate primarily solo-L1 insertions. If transduction propensity differs substantially across elements, the resulting "activity map" may be weighted toward transduction-prone loci rather than reflecting the overall retrotransposition burden.
 3. The observation that L1 hypomethylation is associated with activity is intriguing and could benefit from a more in-depth discussion.
 4. A systematic sensitivity analysis or a comment of how sequencing depth affects detection of source L1 activity, particularly for subclonal events, would strengthen the study. While the reported coverage is moderate, the impact of this limitation on detecting insertions from less active elements is not fully explored.
- Overall, this is a valuable and timely contribution that advances the application of nanopore sequencing to L1 biology.

Reviewer #2

(Remarks to the Author)

Nummi et al report a survey of retrotransposition in a large colon cancer cohort partially using long-read nanopore sequencing. This will be of general interest across colon cancer genomics and retrotransposon genomics. The findings concerning correlates of 5' inversion and transduction length are interesting.

My primary concerns are regarding the approach to TE detection and integration of long and short-read data, particularly as they have different detection characteristics. I am emphasising this in my review because robust insertion calls are a pre-requisite for the study of somatic transductions.

Was the PCR analysis informative regarding false positive rate with respect to somatic vs germline status?

The majority of the normal colon tissues were sequenced via Illumina (12 via Nanopore) so are the other 44 matched to Illumina samples and if so how was this done? If they were compared against short read data, consider a scenario where long-read insertions detected in tumour are marked somatic due to their non-detection in matched normal colon short-read

data. Ideally, what I would want to see is something that integrates both data types or a manual examination of sequence mappings in in the matched-normal short-read data at the sites of insertions detected in the nanopore data (e.g. looking for clipped reads that would indicate an insertion).

Concerning xTea, can you elaborate on 12 xTea runs failing to produce output (p. 19 line 4) - was this a software issue or data issue (e.g. low coverage). If it was the former, I would note that there are other software options available for TE detection (though xTea is generally a good choice).

I didn't see evidence in the supplementary tables that somatic insertions were compared against a database of known polymorphic insertion sites. It is important to consider whether a putative somatic insertion has shown up in a previous study as an additional means of eliminating false-positive somatic insertion calls. I would also note that other nanopore-based surveys of insertion sites exist, which may be useful for comparison.

Furthermore, can somatic insertion sequences (plus flanking sequence) be included as supplemental information? This would be helpful in evaluating the veracity of somatic insertion calls.

Pg 4 Line 15-19, how many L1s are active in the germline but weren't responsible for any somatic insertions in your cohort?

Figure 3a: Previous studies of somatic retrotransposition in cancer have identified few if any full length (6kbp) insertions. It is notable that some are found here.

Figure 5c: is there a correlation coefficient or a significance value associated with the trend line in this panel?

Beyond biological sex, did the authors note any correlation between insertion count or methylation level and patient characteristics? E.g. was patient age, p53 mutation status, or crc subtype (e.g. adeno vs serrated) associated with insertion count or methylation level?

While the data sharing situation around these samples is a bit unfortunate, more could be done to make these data useful to third parties e.g. the availability of bedMethyl files for nanopore data (including allele-specific bedMethyl files from Longshot analysis) would be very useful to some.

Version 1:

Reviewer comments:

Reviewer #1

(Remarks to the Author)

The authors have satisfactorily addressed all of my concerns and corrections. In my view, the manuscript is now suitable for publication in its current form.

Reviewer #2

(Remarks to the Author)

Thank you for your attention and consideration of my concerns regarding this manuscript. I feel the manuscript has been substantially improved and I am satisfied that the false positive rate is likely to be acceptable at this stage.

Dear Dr Belluti, Dr Stortz,

Please find attached our revised manuscript "Nanopore sequencing reveals hidden landscape of short L1 transductions in colorectal cancer" (COMMSBIO-25-6621-T) and our point by point response to the reviewer comments below. We have made multiple small edits to the manuscript and improved data sharing by providing the sites for somatic insertions and methylation levels for the source elements. We expect to have answered thoroughly to all of the points raised by the reviewers and are confident the article is a solid contribution to cancer research.

Below, we have copied the reviewer comments in *grey italic* and our responses are in roman. Quotes from the manuscript are in **bold**.

Reviewers' comments:

Reviewer #1 (Remarks to the Author):

This is a well-written manuscript that leverages one of the key advantages of nanopore sequencing to study L1 transductions in colorectal cancer. The data and conclusions are strong and contribute meaningfully to our understanding of retrotransposition, despite some minor limitations. Nummi et al. convincingly demonstrate that additional sequences generated through L1 transductions have often been missed due to technological constraints. By applying nanopore sequencing to colorectal cancer as a model system, they show that transductions occur more frequently than previously appreciated.

We appreciate the reviewer's kind words and are grateful for their insightful and constructive comments, which we have addressed in detail below.

My comments:

1. It would be helpful to include a discussion of the limitations of nanopore sequencing in this context.

We appreciate the reviewer's suggestion. We agree that the base-calling quality and limited read depth of nanopore sequencing pose certain limitations. Accordingly, we have expanded the discussion to address these points in greater detail.

Specifically, the following statements are included in manuscript:

- High error rate (Page 14, Lines 31-33):

“Leveraging such polymorphisms for source detection would require substantially higher sequencing depth, as nanopore sequencing remains characterized by a relatively high error rate.”

-Low sequencing coverage (clarified from earlier version) (Page 16, Lines 35-37):

“Moreover, sequencing coverage limits our ability to detect subclonal insertions, indicating that L1 activity in certain samples may be present yet remain undetectable without higher sequencing coverage.”

2. The current framework for defining the landscape of “active” L1s relies entirely on transductions, which represent only ~23% of insertion events. This approach introduces a systematic bias by potentially overlooking highly active elements that generate primarily solo-L1 insertions. If transduction propensity differs substantially across elements, the resulting “activity map” may be weighted toward transduction-prone loci rather than reflecting the overall retrotransposition burden.

We thank the reviewer for this insightful comment. The reviewer is absolutely correct that by identifying the sources solely through transduction events, we are not detecting sources for the majority of insertions. This represents an important limitation in the field, as the most established and comprehensive method to infer active source elements remains the detection of transductions. In response to this comment, we have emphasized this limitation more explicitly in the revised manuscript (Page 14, Lines 29-33):

“We detected source L1s by utilizing 3’ transductions, leaving a large portion of somatic activity unassigned. The remaining sources are challenging to identify, as L1 sequences are largely shared among different loci, with only occasional internal polymorphisms. Leveraging such polymorphisms for source detection would require substantially higher sequencing depth, as nanopore sequencing remains characterized by a relatively high error rate.”

3. The observation that L1 hypomethylation is associated with activity is intriguing and could benefit from a more in-depth discussion.

We thank the reviewer for this comment and agree that the association is intriguing. We have now expanded the discussion in the following sentence (Page 16, Lines 1-4):

“Nanopore sequencing provided access not only to the internal L1 sequences but also to information on DNA methylation. Since multiple studies have reported increased L1 activity associated with promoter hypomethylation^{9,26,35,36}, we examined the methylation levels of active source elements.”

9. Tubio, J. M. C. *et al.* Mobile DNA in cancer. Extensive transduction of nonrepetitive DNA mediated by L1 retrotransposition in cancer genomes. *Science* 345, 1251343 (2014).
26. Scott, E. C. *et al.* A hot L1 retrotransposon evades somatic repression and initiates human colorectal cancer. *Genome Res.* **26**, 745–755 (2016).
35. Schauer, S. N. *et al.* L1 retrotransposition is a common feature of mammalian hepatocarcinogenesis. *Genome Res.* **28**, 639–653 (2018).
36. Baba, Y. *et al.* LINE-1 hypomethylation, increased retrotransposition and tumor-specific insertion in upper gastrointestinal cancer. *Cancer Sci.* **115**, 247–256 (2024).

4.A systematic sensitivity analysis or a comment of how sequencing depth affects detection of source L1 activity, particularly for subclonal events, would strengthen the study. While the reported coverage is moderate, the impact of this limitation on detecting insertions from less active elements is not fully explored.

We thank the reviewer for this valuable suggestion. We acknowledge that limited read depth may introduce bias in the detection of source L1 activity, particularly for subclonal events. To address this, we performed a sensitivity analysis to evaluate how sequencing depth affects the detection of somatic insertions and transductions.

As previously reported (Cajuso *et al.*, 2019; Pradhan *et al.*, 2017), the sensitivity to detect subclonal insertions increases with sequencing depth, and this trend appears to continue without a clear upper limit. In our analysis, we modeled the number of somatic insertions as a function of mapped sequence amount (Gbp) using a negative binomial distribution. Figure below (Figure Coverage and insertions) shows 93% high posterior probability interval of number of somatic insertions as function of the mapped sequence

amount, assuming negative binomial distribution of the insertion events. We estimate detecting about 4% more somatic insertions for each gigabase of sequencing (Figure Coverage and insertions).

Figure Coverage and insertions

The number of somatic insertions as a function of the amount of mapped sequence (Gbp), where each dot represents CRC samples (Supplementary Table S3). The shaded region is the 93% high posterior probability range assuming a negative binomial distribution of insertion events.

It is notable that the significant part (we estimate about one fourth) of the estimated increase is due to few samples with fairly typical ~70Gbp of sequence and very high number of detected insertions.

In contrast, the rate of transduction detection, given a somatic insertion, appears independent of sequencing depth. Transduction rate (transductions per insertion) remains stable across varying coverage levels (Figure Coverage and transduction rate). From this, we conclude that once an insertion is detected, the likelihood of identifying a transduction is not influenced by read depth.

Figure Coverage and transduction rate

The somatic transduction rate (transductions/insertions in a sample) as a function of the amount of mapped sequence (Gbp), where each dot represents CRC samples (Supplementary Table S3).

To support transparency, we have added per-sample coverage data to Supplementary Table S3.

Overall, this is a valuable and timely contribution that advances the application of nanopore sequencing to L1 biology.

We are grateful for these insightful and kind comments and hope we answered them satisfactorily.

Reviewer #2 (Remarks to the Author):

Nummi et al report a survey of retrotransposition in a large colon cancer cohort partially using long-read nanopore sequencing. This will be of general interest across colon cancer genomics and retrotransposon genomics. The findings concerning correlates of 5' inversion and transduction length are interesting.

We thank the reviewer for the encouraging comments and are pleased that the findings on 5' inversion and transduction length were found interesting.

My primary concerns are regarding the approach to TE detection and integration of long and short-read data, particularly as they have different detection characteristics. I am emphasising this in my review because robust insertion calls are a pre-requisite for the study of somatic transductions.

We acknowledge the concerns raised here. In this study, we primarily analyzed short-read and long-read sequencing data separately. Only in instances where matched normal samples were unavailable for nanopore sequencing did we use short-read sequenced samples as controls. In these cases, we accounted for the differing characteristics of the sequencing platforms and applied distinct approaches, as detailed later in this response. Additionally, we intentionally compared the sequencing methods to emphasize their distinct characteristics.

Additionally, we wish to clarify the somatic filtering criteria in our previous study (Nummi et al., 2025), which utilized the same sample set and detection methods. This is how we ensured a high true somatic rate.

First, somatic filtering was performed across all available samples ($n = 270$), including both tumor and normal tissues. Importantly, we filtered candidate somatic transposable element TE insertions against all detected insertions, not limited to TEs alone. Furthermore, we restricted the final set of somatic insertions to L1 elements only. This decision was based on our analysis of rare polymorphisms misclassified as somatic events, which indicated that only L1 insertions showed a somatic detection rate exceeding the expected frequency of rare germline variants.

Additionally, we manually inspected 50 somatic L1 insertions in colorectal cancer samples using BasePlayer (Katainen et al., 2018), with matched normal samples sequenced either by Nanopore or Illumina serving as visual controls. We employed different numbers of supporting reads for the different sequencing as detailed in the manuscript. This analysis yielded a true somatic rate of 96% (48/50).

To generalize the validation of somatic calls using Illumina-sequenced normal samples, we assessed the proportion of discordant reads at insertion breakpoints identified in the

Nanopore data. We found that 93% of somatic calls exhibited fewer than 10% discordant reads in the normal samples, in contrast to only 1% of germline calls, supporting a 93% true somatic rate.

Furthermore, a PCR validation for 13 insertions provided clear true somatic status for 10 of them. Three of the insertions showed faint bands in the normals, which raised suspicions of either somatic mosaicism or contamination.

Collectively, these validation approaches consistently demonstrated a high true positive rate (93–96%). To clarify this in the manuscript, we have added the following text to the Methods section (Page 18, Lines 20-24):

“Although the majority of the CRC tumors lacked a nanopore sequenced corresponding normal, we utilized the use of all the other samples as a control. This resulted in somatic L1 calls with 93-96% rate of true somatic events as evaluated by visualization and discordant read counts with short read WGS corresponding normals and targeted PCR experiments²⁸.”

28. Nummi, P. *et al.* Structural features of somatic and germline retrotransposition events in humans. *Mob. DNA* **16**, 20 (2025).

Was the PCR analysis informative regarding false positive rate with respect to somatic vs germline status?

Yes, the PCR analysis was informative regarding the false positive rate in distinguishing somatic from germline insertions. For Nanopore-detected transductions, we validated 18 insertions and observed no amplification in the matched normal samples under identical experimental conditions and primers. For Illumina-detected insertions, 1 out of 13 showed a faint band in the normal sample, which we interpreted as either low-level somatic mosaicism or potential tumor contamination. A similar observation was reported in our previous study (Nummi et al., 2025), where faint bands were also detected in normal tissue.

We added the following section to results to report on the somatic status of the insertions (Page 7, Lines 5-7):

“The nanopore called transductions were shown to be true somatic as corresponding normal samples showed no band in similar conditions. One of the

Illumina calls showed a faint band in the normal, possibly a result of somatic mosaicism or contamination (Supplementary Figure S3)."

The majority of the normal colon tissues were sequenced via Illumina (12 via Nanopore) so are the other 44 matched to Illumina samples and if so how was this done? If they were compared against shortread data, consider a scenario where long-read insertions detected in tumour are marked somatic due to their non-detection in matched normal colon short-read data. Ideally, what I would want to see is something that integrates both data types or a manual examination of sequence mappings in in the matched-normal short-read data at the sites of insertions detected in the nanopore data (e.g. looking for clipped reads that would indicate an insertion).

As discussed above and in our previous study (Nummi et al., 2025) we employed joint evaluation of long-read and short-read data to support the somatic classification of insertions and ensure a high true positive rate. Visualization of 50 insertions resulted in a true positive rate of 96% (48/50). Example figures of visualization are presented in replies further below (Figure Germline insertion and Somatic insertion).

Concerning xTea, can you elaborate on 12 xTea runs failing to produce output (p. 19 line 4) - was this a software issue or data issue (e.g. low coverage). If it was the former, I would note that there are other software options available for TE detection (though xTea is generally a good choice).

The failure resulted from 12 samples with local high coverage merged with technical incompatibilities: as an example in some instances a wtdbg subprocess ran more than 1 week on one insertion and xTea could proceed only after we manually killed the offending subprocess. In other instances xTea produced hundreds of thousands of files per directory causing damage to the file system. These issues could have been worked around manually but doing that would have compromised the comparison to xTea, as published.

We acknowledge that alternative tools for TE detection exist, however, their implementation to our particular data can be a time consuming process that still results in complications with a subset of samples, as observed with xTea. For this reason, we opted to proceed with a subset of samples that successfully ran through xTea, recognizing that this introduces a slight bias in favor of the tool.

I didn't see evidence in the supplementary tables that somatic insertions were compared against a database of known polymorphic insertion sites. It is important to consider whether a putative somatic insertion has shown up in a previous study as an additional means of eliminating false-positive somatic insertion calls. I would also note that other nanopore-based surveys of insertion sites exist, which may be useful for comparison.

We thank the reviewer for this important comment regarding the validation of somatic insertions against known polymorphic sites. To address this, we performed a comparison using the Human Mobile Element Insertion Database (HMEID) (Niu et al., 2022). Of the 349 somatic transductions identified in our study, 4 (1.1%) were located within 200 bp of an L1 element listed in HMEID. For all 1495 somatic L1 insertions, 17 (0.9%) were within this proximity, including the aforementioned transductions.

To further evaluate these insertions, we visualized them using BasePlayer (Katainen et al., 2018) alongside matched normal samples (sequenced with either Nanopore or Illumina). This analysis revealed that 14 of the 17 insertions, including all 4 transductions, were present in the normal samples and thus classified as germline (Figure Germline insertion). The remaining 3 insertions, despite their proximity to known polymorphic sites, were absent in the matched normals and thus retained as somatic (Figure Somatic insertion).

Thus, we came to the conclusion that based on this analysis, 4/349 (1.1%) of somatic transductions and 14/1495 (0.9%) of all somatic L1 insertions were germline and present in the HMEID database. These rates are well within our estimates of the true somatic rate (93%-96%).

Figure Germline insertion

An example of visualization of an insertion detected as somatic, but a match in the HMEID database and visualization proved to be germline. The upper panel shows the insertion in nanopore sequenced tumor, and the middle panel in Illumina sequenced. The bottom panel is the corresponding normal sequenced with Illumina, showcasing the discordant and split reads characterizing the insertion.

Figure Somatic insertion

An example of visualization of an insertion detected as somatic, that matched in the HMEID database, however a visualization indicated the sample to be somatic. The upper panel shows the insertion in nanopore sequenced tumor, and the middle panel in Illumina sequenced. The bottom panel is the corresponding normal sequenced with Illumina, lacking the characteristic read support of insertions, with only sporadic background discordant reads present.

Furthermore, can somatic insertion sequences (plus flanking sequence) be included as supplemental information? This would be helpful in evaluating the veracity of somatic insertion calls.

We appreciate the reviewer's suggestion to include somatic insertion sequences for further evaluation. In response, we have added all detected somatic L1 insertion sequences along with their annotations as supplementary files, hosted on Zenodo.

Given the sensitive nature of genomic data, we have masked most of the flanking sequences to protect participant identity. However, we retained 50 bp of flanking sequence on both sides of each insertion to allow for contextual evaluation. Importantly, we chose to mask rather than remove the remaining flanking sequence to preserve the integrity of the annotation coordinates.

The link is provided in data availability (Page 21, Lines 14-16):

“Large additional datasets are available in Zenodo <https://doi.org/10.5281/zenodo.17293682>. They include somatic insertion sequences along with their annotations. Additionally, it contains 5' methylation of reference L1-HS in our samples.”

Pg 4 Line 15-19, how many L1s are active in the germline but weren't responsible for any somatic insertions in your cohort?

In our study, we identified 25 L1 source elements with detectable germline transductions. Of these, 7 elements did not contribute to any somatic transductions in our cohort.

This information can be inferred from Supplementary Table S1, which lists all source elements along with their germline and somatic activity.

Figure 3a: Previous studies of somatic retrotransposition in cancer have identified few if any full length (6kbp) insertions. It is notable that some are found here.

We thank the reviewer for highlighting this important point. Indeed, somatic full-length L1 insertions (~6 kbp) are rare. In our cohort, we identified five such full-length somatic insertions. Of these five, three were partnered transductions, indicating that the somatic full-length L1 elements can be transposition-competent.

Inspired by this comment, we have added the following clarification to the manuscript (Page 11, Lines 12-16):

“The second unusual transduction event is a somatic full length insertion 6p12.3-1. Full length somatic L1s (>5990 bp) are rare events, as we report 5 of them, constituting 0.3% of all somatic L1s (5/1495). One of them, the insertion in 6p12.3-1 gave rise to five downstream somatic insertions in the corresponding tumor, being responsible for 5/68 of the somatic insertions in the sample.”

Figure 5c: is there a correlation coefficient or a significance value associated with the trend line in this panel?

We thank the reviewer for pointing this out. Yes, there is a correlation coefficient and significance value associated with the trend line in Figure 5c. We have now added the following clarification to the manuscript (Pages 13-14, Lines 21-1):

“Samples with transductions exhibited lower average methylation than those without, although variation was present among transduction-negative samples (Fig. 5c) ($r = -0.274$, 95% CI $[-0.501, -0.012]$, $p = 0.041$).”

Beyond biological sex, did the authors note any correlation between insertion count or methylation level and patient characteristics? E.g. was patient age, p53 mutation status, or crc subtype (e.g. adeno vs serrated) associated with insertion count or methylation level?

We thank the reviewer for this insightful question. The relationship between retrotransposition and patient characteristics has been a focus of our earlier work. (Cajuso et al., 2018). In the study, we analyzed a cohort of 202 colorectal cancer (CRC) samples and investigated correlations between insertion count and several clinical and molecular features, including patient age, p53 mutation status, tumor location, microsatellite instability (MSI) vs. microsatellite stability (MSS), allelic imbalance (AI), CpG island methylator phenotype (CIMP), and patient survival. We found significant correlation with CIMP, AI and survival.

In the current study, methylation is introduced as a novel feature through long-read sequencing. However, due to the limited sample size ($n = 56$), of which 36 samples overlap with the previous study, we chose not to repeat statistical analyses for these clinical variables. We reasoned that the reduced statistical power would likely prevent meaningful conclusions from being drawn.

Regarding biological sex, we examined its relevance in the context of the highly active L1 element Xp22.2-1, which is located on the sex chromosome X. This gave a robust reason to study it in this context.

While the data sharing situation around these samples is a bit unfortunate, more could be done to make these data useful to third parties e.g. the availability of bedMethyl files for nanopore data (including allele-specific bedMethyl files from Longshot analysis) would be very useful to some.

We agree with the reviewer's statement and appreciate the suggestion to improve data accessibility. In response, we have produced a file available on Zenodo that reports methylation levels of L1 elements across all samples used in the study. This dataset provides a useful overview of L1 methylation patterns and complements the somatic insertion data.

The link is provided in data availability (Page 21, Lines 14-16):

“Large additional datasets are available in Zenodo <https://doi.org/10.5281/zenodo.17293682>. They include somatic insertion sequences along with their annotations. Additionally, it contains 5' methylation of reference L1-HS in our samples.”

References

- Cajuso, T., Sulo, P., Tanskanen, T., Katainen, R., Taira, A., Hänninen, U. A., Kondelin, J., Forsström, L., Välimäki, N., Aavikko, M., Kaasinen, E., Ristimäki, A., Koskensalo, S., Lepistö, A., Renkonen-Sinisalo, L., Seppälä, T., Kuopio, T., Böhm, J., Mecklin, J.-P., ... Aaltonen, L. A. (2019). Retrotransposon insertions can initiate colorectal cancer and are associated with poor survival. *Nature Communications*, *10*(1), 1–9.
- Katainen, R., Donner, I., Cajuso, T., Kaasinen, E., Palin, K., Mäkinen, V., Aaltonen, L. A., & Pitkänen, E. (2018). Discovery of potential causative mutations in human coding and noncoding genome with the interactive software BasePlayer. *Nature*

Protocols, 13(11), 2580–2600.

Niu, Y., Teng, X., Zhou, H., Shi, Y., Li, Y., Tang, Y., Zhang, P., Luo, H., Kang, Q., Xu, T., & He, S. (2022). Characterizing mobile element insertions in 5675 genomes.

Nucleic Acids Research, 50(5), 2493–2508.

Nummi, P., Cajuso, T., Norri, T., Taira, A., Kuisma, H., Välimäki, N., Lepistö, A., Renkonen-Sinisalo, L., Koskensalo, S., Seppälä, T. T., Ristimäki, A., Tahkola, K., Mattila, A., Böhm, J., Mecklin, J.-P., Silli, E., Pasanen, A., Heikinheimo, O., Bützow, R., ... Aaltonen, L. A. (2025). Structural features of somatic and germline retrotransposition events in humans. *Mobile DNA*, 16(1), 20.

Pradhan, B., Cajuso, T., Katainen, R., Sulo, P., Tanskanen, T., Kilpivaara, O., Pitkänen, E., Aaltonen, L. A., Kauppi, L., & Palin, K. (2017). Detection of subclonal L1 transductions in colorectal cancer by long-distance inverse-PCR and Nanopore sequencing. *Scientific Reports*, 7(1), 14521.